# Intermediate Filaments from Tissue Integrity to Single Molecule Mechanics

**DOI:** 10.3390/cells10081905

**Published:** 2021-07-27

**Authors:** Emma J. van Bodegraven, Sandrine Etienne-Manneville

**Affiliations:** Cell Polarity, Migration and Cancer Unit, Institut Pasteur, UMR3691 CNRS, Equipe Labellisée Ligue Contre le Cancer, F-75015 Paris, France; emma.van-bodegraven@pasteur.fr

**Keywords:** cytoskeleton, mechanics, resilience, rigidity, stiffness, elasticity, viscosity, rod domains, coiled-coil region

## Abstract

Cytoplasmic intermediate filaments (IFs), which together with actin and microtubules form the cytoskeleton, are composed of a large and diverse family of proteins. Efforts to elucidate the molecular mechanisms responsible for IF-associated diseases increasingly point towards a major contribution of IFs to the cell’s ability to adapt, resist and respond to mechanical challenges. From these observations, which echo the impressive resilience of IFs in vitro, we here discuss the role of IFs as master integrators of cell and tissue mechanics. In this review, we summarize our current understanding of the contribution of IFs to cell and tissue mechanics and explain these results in light of recent in vitro studies that have investigated physical properties of single IFs and IF networks. Finally, we highlight how changes in IF gene expression, network assembly dynamics, and post-translational modifications can tune IF properties to adapt cell and tissue mechanics to changing environments.

## 1. Introduction

Tissue integrity, which is necessary for all metazoan life, relies on the ability of cells to adapt their morphology, their interactions, and their function to the conditions of their environment. The cytoskeleton, including actin microfilaments, microtubules, and intermediate filaments (IFs), form essential intracellular networks which support cell shape, cell adhesions and are indispensable for most cellular functions. While the role of actin in cell morphology, motility, and contractility has been extensively studied and the contribution of microtubules to intracellular trafficking, cell polarity, and adhesion dynamics is now well understood, the role of IFs in cell functions and tissue integrity remains unclear. This is partly because, in contrast to the ubiquitously expressed actin and tubulin, IF protein expression varies between cell types and tissues, and IF protein levels can represent anything from 0.3 to 85% of total protein levels in the cell [1,2]. Despite a high level of shared structural features between cytoplasmic IFs such as a head, rod, and tail domain, the more than 70 known IF genes create highly specialized, cell-type-specific networks of polymeric filaments. IFs are subdivided into five subtypes depending on small structural differences, their modes of assembly, and their expression pattern [3]. GFAP, vimentin, synemin, and nestin form the IF network in glia, neurofilaments in neurons, desmin and syncoilin in muscles, keratins in skin, and vimentin in mesenchymal cells. Consequently, the depletion of single IF genes does not always lead to severe phenotypes. However, in humans, mutations in IF genes give rise to a large diversity of diseases commonly characterized by the altered integrity of specific tissues [4]. The lack of associated molecular motors and well-characterized regulators of the assembly/disassembly dynamics further distinguishes IFs from actin and microtubules. These characteristics are probably responsible for our late understanding of IF functions at the cellular and tissue level. Only the painstaking studies of each type of IF’s structural and mechanical properties and their integration at the network, cellular and tissue levels are slowly unraveling the contribution of IFs to the physiology and pathology of multicellular organisms.

## 2. Intermediate Filaments as Key Players in Tissue and Cellular Mechanics

### 2.1. Intermediate Filaments, Guardians of Tissue Integrity

Disease-causing mutations in IFs showcase the contribution of IFs to tissue resilience against mechanical stress [3,5]. This is particularly well illustrated in skeletal and cardiac muscles, where mutations in the muscle-specific IF protein desmin lead to various diseases, collectively called desminopathies [6]. For instance, a desmin mutation (D399Y) that causes muscle weakness and respiratory failure decreases myoblast’s ability to elongate and spread during uniaxial cyclic elongation [7]. Similarly, mutations in keratin 5 and keratin 14 identified in early studies cause epidermolysis bullosa simplex, a disease that causes skin blistering [8,9,10,11]. More recently, the asymmetric inheritance of keratin was shown to control the specification of the placenta and the embryo, thereby demonstrating the critical role of keratins during early development [12]. The trophectoderm cells inherit keratin 8 and 18 (K8/18), which assemble in a dense keratin network to generate mechanical stability required for the embryo cavitation (Figure 1A). In the absence of K8/18, the embryo cavitates to form the blastocyst; however, the volume of the blastocyst is decreased while its surface curvature increases, indicating a lower apical tension and reduced stiffness. The abnormalities are reversed by the re-expression of keratin [12]. The fundamental role of IFs in muscular and epithelial tissues, which face frequent mechanical stresses both during development and in adults, has led to the hypothesis that IFs participate in the control of cell and tissue mechanics.

### 2.2. Intermediate Filaments, Guardians of Cell Integrity

At the cellular level, studies have progressively confirmed that IF’s function in tissue integrity relies on their contribution to cell resilience under both mechanical stretching and compression. Indeed, the depletion of keratin or vimentin increases the deformability of stretched cells [13,14] (Figure 1B,C). Importantly, the loss of vimentin also decreases the viability of cells submitted to stretching [15] (Figure 1C). The role of IFs in cellular resistance to deformation by compressive forces appears to be cell-type specific. Under compression, both the depletion of vimentin in human mesenchymal stem cells [16] and the overexpression of vimentin in amoeboid cancer cells [17] reduce cell deformation. This discrepancy may be due to a different initial level of IF protein expression optimized in a cell-type-specific manner to provide cells with different mechanical properties. Alternatively, the difference may result from a cell-type-specific composition of the cytoskeletal network. For example, the depletion of vimentin in compressed highly contractile cells might have a different effect on their deformability compared to depletion in cells with lower contractility. Taken together, these studies show how IFs contribute to cell resilience by limiting cell deformation under mechanical stress and allowing stretched or compressed cells to recover their initial shape without any damage. However, they also suggest that the mechanical resilience of a given cell type in a specific situation may require an optimal level of expression of a specific IF type.

The IF network can also reorganize in response to mechanical stresses [18,19,20,21,22]. Rearrangement of the network is nicely illustrated in epithelial domes. In this three-dimensional in vitro epithelial sheet model, cells undergo extreme deformations while the tension across the epithelial sheet is maintained constant [23]. In fully expanded domes, extremely stretched cells coexist with cells that barely change their shape. The stretched cells are characterized by the formation of unusually straight bundles of keratin IFs, which extend from the nucleus to the plasma membrane. Laser ablation of these IF bundles results in a rapid increase in the cell area [23] (Figure 1D). This suggests that keratin IFs in highly stretched cells are load-bearing elements that maintain the reversibility of cell shape after large deformations. At the cellular scale, cell stretching by hypo-osmotic stress partly depolymerizes vimentin and nestin filaments and redistributes them throughout the cells [21]. This rearrangement is essential to cell survival after hypo-osmotic shock, confirming the contribution of IFs to cell mechanical resistance.

### 2.3. Intermediate Filaments Adapt Cell Mechanics to Cell Behavior

Accumulating evidence shows that IF networks are rearranged when cells adapt to external mechanical challenges. Cells can generate contractile or pushing forces to reshape in order to accomplish specific functions such as cell division or migration. To divide, cells must actively generate forces to accommodate their shape changes and overcome mechanical constraints by the surrounding tissue. The IF network reorganizes extensively not only to adapt to but also to promote these changes [24,25,26,27]. As cells round up to facilitate the accurate positioning of the spindle and the correct segregation of chromosomes, cortical tension generated by the actin cortex increases. In HeLa Kyoto cells, vimentin IFs contribute to this increase in cortical tension by relocalizing to the cortex, where they interact with actin to control actin organization [28,29] (Figure 1E). In confined environments, the loss of vimentin becomes detrimental to the segregation of chromosomes, and chromosome lagging is often observed [28]. However, besides vimentin IFs, HeLa cells also express keratins, which also reorganize during mitosis and affect the organization of vimentin. Indeed, when cells express nestin, the reorganization of vimentin during cell division is different. In nestin-expressing ovary (CHO) cells, C6-2 glioma, BHK-21 fibroblast, and cerebellar ST15A cells, the vimentin network disassembles at the cleavage furrow and does not localize to the cortex [30,31]. It thus seems that the reorganization of IFs that accompanies the changes in cell mechanics depends on IF proteins that are expressed. The cell-type-specific composition of the IF network needs to be taken into account in future investigations.

Migration in confined environments requires resilient mechanical support to allow cell deformation but prevent damage as they pass through complex environments. IFs appear to provide the essential mechanical support. Keratin knock-out (KO) keratinocytes migrate faster when squeezing through small pores in a Boyden chamber assay. However, they also frequently rupture and die [13] (Figure 1F). Similarly, vimentin depletion facilitates the migration of MEFs (mouse embryonic fibroblasts) in confined environments such as microchannels, collagen gels, and small pores [32,33] at the cost of nuclear alterations, nuclear envelop ruptures, and blebs [32] (Figure 1F). These observations were also confirmed in studies investigating the amoeboid migration of melanoma cancer cells [17]. Both keratin and vimentin networks appear to provide mechanical support to protect the nucleus against excessive deformations and maintain nuclear homeostasis during confined cell migration [32,33,34]. However, the switch from keratin to vimentin expression observed during the epithelial-to-mesenchymal transition (EMT) suggests that the two IF networks differentially contribute to the cell’s mechanical properties [35,36,37,38]. The microinjection of purified vimentin into MCF-7 epithelial cells changes the cell shape to a mesenchymal cell morphology [36]. Whether this effect is solely due to the mechanical functions of IFs or also reflects their role in intracellular signaling and cell motility remains unclear. While IFs in general, and the organization of the cytoplasmic IF network in particular, are essential to provide cell mechanical resilience, it is tempting to speculate that the mechanical specificity of each type of IF participates in cell-type-specific mechanics. Therefore, the control of IF protein expression may be central to the acquisition of cell-type-specific mechanical behavior adapted to the properties of their microenvironment and the modifications of cell behavior observed in pathological situations.

## 3. The Intermediate Filament Network Forms an Intracellular Structural Scaffold

The function of IFs in cell and tissue resilience is based on their contribution to the cell’s ability to resist deformation under mechanical stresses or to deform and recover their original shape when the constraints are lifted (Figure 2A). In physical terms, cells behave both as an elastic solid and as a viscous fluid and are therefore considered viscoelastic [39,40]. Elasticity makes a cell resistant to deformations comparable to the resistance of a spring, in which the energy that induces the deformation is stored independently of time (Figure 2B). In contrast, viscosity makes material flow as a fluid. The resistance of the material then depends on the rate of deformation, and the energy causing the deformation is not stored but dissipated (Figure 2B). Numerous techniques measure cell responses to applied stresses and help to quantify their elastic (G’) and viscous (G’’) modulus, both measured in Pa. The term stiffness is often used as an overarching term to indicate the resistance of material against deformation (Figure 2C). To determine the contribution of IFs to cell mechanical properties, several biophysical approaches have been used (Figure 2D,E). Single cell rheology techniques such as microplate rheology (Figure 2E) assess whole-cell mechanics, while local mechanical properties can be obtained using optical magnetic twisting cytometry (OMTC) with different embedment depths of beads (Figure 2E), atomic force microscopy (AFM) at various indentation depths (Figure 2F), and passive and optical tweezer microbead rheology (Figure 2G). Altogether, these biophysical approaches have shed light on the contribution of IFs in the different mechanical properties of cells.

### 3.1. The IF Network Contributes to Cell Stiffness

In an early study, twisting fibroblasts with integrin-bound magnetic beads showed that the loss of vimentin decreases cell stiffness [41] (Figure 2D), hinting at the contribution of IFs to cell resistance to deformation. This was confirmed for desmin in myoblasts using single-cell rheology (Figure 2E) [42]. The desmin mutant E413K either causes a collapse of the existing desmin network or integrates into the network. Cell softening is observed for cells in which the IF network collapses, whereas cells stiffen when the desmin mutant integrates into the endogenous network or when desmin WT is overexpressed [42]. More local probing, using AFM (Figure 2F), shows that the loss of keratin or vimentin generally reduces cell stiffness [11,14,19,32,33,43]. However, while keratin similarly impacts the entire cell [11,43], vimentin depletion acts more specifically in perinuclear regions [32,33]. Distinguishing cortical and cytoplasmic mechanics with AFM probing either the cell surface or deeper into the cell does show that vimentin contributes to both cortical and cytoplasmic stiffness of MEFs [44,45]. Optical magnetic twisting cytometry (OMTC) (Figure 2D) confirmed the contribution of vimentin to cortical stiffness, but only at large deformations [46]. In contrast, expression of a desmin mutant (E413K) or WT desmin only affects cell stiffness when fibronectin-coated beads are embedded deep enough to reach the desmin network beyond the cell cortex [42]. The distinct contribution of vimentin and desmin to cell stiffness may be due to differences in the mechanical properties or the intracellular organization of each network. It may also reflect variations in the contribution of the actin cytoskeleton to cell mechanical properties; the actin-rich cortex would dominate cortical mechanics under small deformations, while the IF network could appear as the main contributor to cell mechanical properties at large deformations. A systematic and direct comparison of the mechanical contribution of each IF type in the presence or absence of actin would allow us to determine how the composition of the IF network influences cell mechanical properties.

### 3.2. The IF Network Contributes to Cytoplasmic Strength and Toughness

Intracellular mechanics can be further characterized by analyzing the movement of microbeads larger than the mesh size of the cytoskeleton and physically constrained within the network. Using optical tweezers to actively move the beads confirms the softening of the cytoplasm in vimentin-depleted MEFs [47]. Dragging intracellular beads at a constant speed increasingly deforms the cell, while the energy required for the deformation is stored (Figure 2G). The stored energy reaches a maximum, after which, the material yields and changes its behavior from elastic to plastic. This maximum, or peak force, characterizes the cytoplasmic strength, indicating the threshold above which cells cannot recover their initial shape after deformation (Figure 2G). Vimentin depletion decreases the cytoplasmic strength in MEFs [15]. The distance the bead can move until this peak force is reached, defined as stretchability, and the total energy the cell withstands without breaking, defined as cytoplasmic toughness, are also decreased in vimentin-depleted cells [15] (Figure 2G). In contrast, vimentin overexpression increases cytoplasmic strength, stretchability, and toughness.

IF contribution to cell viscosity is evaluated using force–displacement curves generated by AFM. Vimentin depletion increases cell viscosity [45]. In contrast to a purely elastic material, the deformation of a viscous material increases with time, as a constant force is applied and the energy responsible for the deformation is dissipated (Figure 2B). In cells, energy dissipation inversely relates to the restoring force and cytoplasmic toughness. WT and vimentin-depleted MEFs appear both viscous and elastic. Their cytoplasmic toughness and cytoplasmic strength depend on the deformation rate [15]. However, when actin and microtubules are removed to generate ghost cells in which only vimentin remains, the rate-dependency is not observed. While increasing the duration of deformation reduces the restoring force in WT and vimentin-depleted cells, this reduction is not observed in ghost cells. These observations indicate that IFs form an elastic meshwork with negligible viscosity [15], while actin and microtubule networks strongly contribute to cell viscosity [45].

Multiple cycles of force application and relaxation, i.e., loading and unloading tests, better reveal the IF network’s high elasticity and low viscosity. In WT and vimentin-depleted cells, the restoring force decreases with each cycle, reflecting the dissipation of energy [15]. The same repeated loading and unloading cycles do not change the force–displacement curves obtained in ghost cells. After ten cycles, the force–displacement curve plateaus for both WT and vimentin-depleted cells, but the elastic energy remains much higher in WT than in vimentin-depleted cells. In the absence of vimentin, the resistance force strongly decreases as the cyclic loading disrupts most cytoskeletal structures [15]. Vimentin IFs appear to participate in the resilience of the cytoplasm by contributing to its elastic properties, while the other cytoplasmic components essentially contribute to the viscous response. However, the cytoplasmic toughness is larger in WT cells than in vimentin-depleted cells and ghost cells, suggesting that the interplay between vimentin, actin, and microtubule networks is needed to absorb large amounts of energy [15]. Extending these experiments to other cell types will tell us whether the observations obtained from vimentin-expressing fibroblasts can be generalized to different types of IFs.

If the composition of the IF network reflects the mechanical properties of the cell’s microenvironment, whereas actin and microtubule properties remain unchanged, one wonders how IF composition affects the cytoskeletal interplay and the global mechanical behavior of the cytoskeleton. To this end, signaling properties of specific IFs [18,48] may be involved to control the organization and function of the actin and microtubule network in a coordinated manner.

### 3.3. Regulation of the IF Network Allows Local Adaptation of Cell Deformability

While IFs globally contribute to cytoplasmic mechanics at the whole-cell scale, regional differences have been revealed by the analysis of the passive movement of intracellular beads in migrating alveolar epithelial cells [49]. In ghost cells lacking both actin and microtubules, the peripheral keratin network is characterized by a large mesh size and a low elastic modulus, probably facilitating the deformation of the leading edge. When cells are submitted to shear stress, the mesh size of the keratin network decreases, and the elastic modulus increases at the cell periphery. The perinuclear area always presents a high elastic modulus corresponding to a small keratin networkf mesh size. This perinuclear organization likely acts as mechanical protection for the nucleus by limiting nuclear deformations [49]. These observations also suggest that spatial and temporal control of IF network organization actively contributes to the local adaptation of cell mechanics. External cues and intracellular signaling, which control the organization of the network, and thereby its mechanical properties, are only beginning to be deciphered. They certainly involve post-translational modifications and, in particular, the phosphorylation of IF proteins [50].

## 4. Mechanical Properties of IF Networks and Single Filaments In Vitro

In contrast to actin microfilaments and microtubules, which interact with motors such as myosin, dynein, and kinesin, IFs do not bear molecular motors to provide mechanical forces. Instead, it is the fundamental structure of IFs that is at the heart of their mechanical properties (Figure 3). In vitro studies of reconstituted IF networks and single filaments have recently shed light on the structural bases of IF mechanical properties. In this chapter, we recapitulate the results obtained with oscillatory shear rheology experiments on in vitro assembled IF networks which demonstrate their high elastic properties. We discuss how the mechanical properties of IF networks rely on inter-filament interactions (Figure 4) and single filament mechanics (Figure 5) based on biophysical in vitro measurements.

### 4.1. Intermediate Filaments Form Strong Elastic Networks

Oscillatory shear rheology experiments have been used to impose a controlled deformation on in vitro assembled vimentin and keratin networks [58,59] (Figure 4). De novo assembled networks have high elastic moduli, which increases up to 60 min after assembly [60,61]. Both networks are essentially elastic at low strains, with the G’ of mature networks 5 to 10 times higher than their viscous modulus G’’ [58,61,62,63]. Within low stain ranges, increasing strain does not affect G’ and G’’. The corresponding G’ is called the plateau modulus or G^0^ [58,61,62,63,64,65,66,67,68], which is unique to IF networks (Figure 4). Further increasing strain increases G’ and G’’; the network strain-stiffens (Figure 4). In contrast to actin networks which strain-stiffen at small strains of a few percent, IF networks begin stiffening in response to 10 to 20% strain [59,69,70,71]. At 80% strain, G’ can increase by more than an order of magnitude [64] (Figure 4). This strain-stiffening is entirely reversible [72]. All IF networks appear to share high elasticity and strain-stiffening properties at high strains, which distinguish them from other cytoskeletal networks. These mechanical properties of the IF network reflect the properties of interactions between individual filaments and the intrinsic physical properties of isolated filaments themselves.

### 4.2. Inter-Filament Interactions Contribute to the Mechanical Properties of IF Networks

The IF network’s high elasticity and characteristic G^0^ result from inter-filament interactions, i.e., crosslinks, that maintain IF proteins within single filaments in a stretched conformation [61,62] (Figure 4). Maintaining single IF proteins in a stretched conformation requires energy [73], which is provided by the hydrophobic and electrostatic inter-filament interactions. For both keratin and vimentin IFs, neutralizing the attraction with a non-ionic surfactant allows individual filaments to return to their favorable thermal equilibrium state, and G^0^ drops [61,62] (Figure 4). Since tailless mutants do not affect G^0^ [62], these interactions probably occur at the filament rod domains [62,63,74,75].

The strain-stiffening behavior of the IF network, observed at high strains, results from further filament stretching between crosslinks (Figure 4) and is completely suppressed by the addition of a non-ionic surfactant [62]. Strain-stiffening is lost in networks formed by tailless filaments [62,63,74,75], showing that the filament C-terminal tails are responsible for the strong, attractive interactions necessary to withstand high stresses [62]. For instance, the characteristic side arms of neurofilament proteins are thought to be involved in the crosslinking of the network and provide resistance at large deformations [76].

The strain-stiffening of the network is limited by the rupture of the network [60,62,63,65] (Figure 4). The stress at which the network ruptures, called the yield stress, is much higher for IF than for actin networks. When the yield stress is reached, a softening of the network is observed, which depends on the nature and degree of inter-filament interactions. The softening of vimentin IF networks is transient, which also distinguishes them from actin networks [77,78], and indicates that filaments are not permanently fractured [72]. Strengthening attractive interactions by the addition of divalent cations [61,63,64,65,68] or permanently crosslinking the network [72] increases the yield stress, suggesting that the softening of IF networks is due to the loss of interactions between IF proteins [72] (Figure 4). The softening of vimentin IF networks is also loading-rate-dependent [72] (Figure 4). At slow loading rates, the disruption of transient C-terminal inter-filament interactions counteracts the strain-stiffening response [72]. Here, the mechanical response is dominated by the disruption of crosslinks which occurs before or at the same time as the stiffening of the network and results in a low yield stress (Figure 4) [72]. At fast loading rates, the strain-stiffening precedes the disruption of crosslinks, and the yield stress is much higher (Figure 4) [72]. Similarly, transient interactions between neurofilament side arms are thought to be responsible for the reorganization of the network following disruption by large prolonged strains [76]. At fast deformations, these interactions provide the IF network with mechanical resilience [76].

Interestingly, the yield stress lies within the range of contractile forces exerted by cells. In fact, strain-stiffening cellular responses are observed in cells in which actin and microtubules are removed, but not in vimentin-depleted cells [15]. Cell contractile behavior may affect the IF network and cell mechanics. Cell deformation in response to a fast contraction may be limited by the fast strain-stiffening of the IF network. Inversely, a slow actin–myosin-driven contraction would modify interactions between IFs and remodel the IF network. Thus, the slow disruption of inter-filament interactions probably also contributes to the dissipation of energy, which has mainly been attributed to the conformational changes of individual IF proteins [72].

### 4.3. Single Intermediate Filaments Are Flexible and Stretchable

Single IFs are much more flexible than actin microfilaments and microtubules in vitro [58,79,80] (Figure 3). The flexibility of filaments is quantitatively characterized by their bending rigidity. The bending rigidity can be derived from the filament persistence length, which is the length over which the filament direction does not change (Figure 3). The persistence length of vimentin is similar in cells and in vitro. However, it is slightly higher when in vitro filaments are adhered to substrates [81]. Interestingly, the depolymerization of both actin and microtubules increases IF persistence length measured in cells. This suggests that when IFs bear all the mechanical load, they tend to stiffen and possibly play a protective mechanical role when the other cytoskeletal elements are absent or altered [81].

The flexibility of IFs varies with their composition. For instance, neurofilaments are softer than other IFs, which may be relevant to their expression in soft brain tissue [82]. The persistence length can also vary along the length of a single filament [83]. These variations may be due to local changes in the subunit composition. The ratio between neurofilament light chain and heavy chain influences the persistence length of the filament, possibly because side arms of neurofilament proteins can affect the interaction between subunits [82]. Whether cells can locally and dynamically modify the composition of IFs remains to be investigated. This would necessitate the local recruitment of specific IF proteins and require a controlled exchange of subunits within preexisting filaments or local regulation of the polymerization.

A second specific characteristic of IFs is their high stretchability and strong resistance to breakage. In vitro, single IFs can be stretched about 250% before breakage and can even reach 350% for desmin filaments [84,85]. For low strains up to 100%, a steep increase in force is observed with increasing strain [85,86,87,88]. The force–strain curve plateaus for intermediate strains, and at higher strains, the force further increases, indicating strain-stiffening of the filament [88] (Figure 5A). X-ray experiments [89] and mathematical modeling approaches suggest that the steep linear increase in force at low strains results from the elastic stretching of the coiled-coil α-helical domains [88,90,91] (Figure 5A). Further stretching of the α-helical domains leads to their transition to β-sheets, reflected by the force plateau at intermediate strains. The stiffening of the filaments at higher strains corresponds to the necessity to exert higher forces to extend β-sheets further (Figure 5A). This conformational change and the stretching of single filaments in a β-sheet conformation between filaments in IF networks are essential to the high G^0^ and strain-stiffening of the network, respectively (Figure 4).

As viscoelastic material, the IF’s mechanical response is loading-rate-dependent. Slow deformations do not cause stiffening of vimentin IFs before 200% strain, but fast deformations induce stiffening at 50% strain [88] (Figure 5A). This physical behavior of IFs has led to the description of IFs as safety belts.

Upon repeated loading and unloading cycles, filaments progressively soften, indicating a dissipation of energy [73] (Figure 5B). The filament recovers its initial length between consecutive cycles, but the energy needed to stretch the filament a second time is always lower, no matter the time spent between the first and the second cycle [92]. This suggests that alterations in mechanical properties following filament stretching are not reversible within reasonable time scales, a signature of a tensile memory. A potential explanation of IF tensile memory is that the α-helical rod domain of each monomer reacts independently to the applied load [73] (Figure 5B). During unloading, some monomers remain in the β-sheet conformation, forming a ULF with a mix of proteins in α-helix and β-sheet conformations (Figure 5B). With each loading and unloading cycle, an increasing number of α-helices transit to β-sheets. The remaining α-helices, which are shorter than β-sheets, determine the filament length and thus primarily experience the force applied during re-stretching. As long as some monomers are still in an α-helical state, the filament returns to its initial length, but as the number of α-helices decreases, the force needed to achieve the same strain decreases. The softening of the filament and apparent dissipation of energy are explained by the higher energy required to keep the monomer in a β-sheet state within the relaxed filament. Part of the energy used to stretch the filament is stored in the monomers in a β-sheet state and is dissipated when β-sheet elements return to the α-helical conformation upon relaxation [73].

In contrast, the initial mechanical properties of in vitro assembled IF networks are completely recovered after repeated loading [72], and no evidence for tensile memory in IF networks has been reported. This makes the applicability of these observations to IF networks in vitro and in cells debatable. Interestingly, when single filaments are permanently crosslinked, IF stiffness is totally recovered after stretching, suggesting that IF conformational changes are reversible if the degrees of freedom are limited [92]. To explain the inability of IFs to recover their initial mechanical properties, a third conformational state has been introduced, which is disordered, elongated, and disabled to return back to the α-helix conformation [92] (Figure 5C). IF protein crosslinking could prevent the disordered conformational state and help the full recovery of α-helices. It is interesting to speculate that inter-filament interactions in IF networks limit the degrees of freedom of IF monomers to a similar extend. In cells, not only crosslinking but also subunit exchanges or disassembly and assembly of the filaments can be achieved. Up to now, the contribution of these various mechanisms to the recovery of IF mechanical properties, and thereby to the maintenance of cell integrity, remains unclear.

Recently, non-vibrational spectroscopy experiments have shown the α-to-β sheet conformational change in living cells under different cellular tension [93]. In relaxed cells, vimentin is in α-helical conformation. In contrast, when cells are under tension, β-sheet conformations are mostly observed [93]. This implies that single filaments display similar mechanical behavior in cells as in vitro. Thus, mechanical properties observed in vitro do give us insight into the role of the IF network in cell mechanics.

## 5. Modulation of Intermediate Filaments Influences Cell Mechanics

Each tissue and organ display specific mechanical properties. Tissue stiffness can vary from very soft, like the brain, to stiff, such as bone. As cells differentiate within tissues or change location during development and in adulthood, they must adapt their mechanical properties to their environment. This is most profound in mechanically challenging tissues such as the skin or the muscles, where IFs are not surprisingly essential for the maintenance of mechanical integrity. Since IFs appear as key players in cell mechanics, one can wonder whether and how changes in expression can modulate IF networks to adapt their mechanical properties to changing environments.

### 5.1. Intermediate Filament Mechanics Can Be Tuned

One way to modify the IF network is to change its composition. As cells undergo epithelial-to-mesenchymal transition (EMT) and start invading mesenchymal tissue, vimentin expression dramatically increases, while keratin levels tend to decrease. The force–strain behavior of single vimentin and keratin filaments in vitro is different and indicates that higher forces are needed to deform vimentin than keratin [94]. Increased initial stiffening for vimentin IFs can be attributed to a stronger attraction and increased lateral coupling between vimentin subunits (Figure 6A,B). Mathematical modeling indicates that the lateral coupling in vimentin is so strong that filament elongation can only occur when all parallel α-helices within a ULF are unfolded (Figure 6C). For keratin, due to weaker lateral interactions, the filament can elongate as soon as one subunit has unfolded [94] (Figure 6C). Furthermore, in contrast to vimentin IFs, keratin filaments are not compacted after assembly due to opposing charges of amino acids between neighboring dimers [58] (Figure 6B). Filament compaction can increase the lateral coupling strength of vimentin IFs and contribute to their increased initial stiffening.

The composition of IF networks can also impact their sensitivity to intracellular signals. The assembly of keratins into crosslinked IF networks relies on the high fraction of hydrophobic amino acids in their rod and tail domains [62], while vimentin networks mostly rely on the negatively charged amino acids to engage in electrostatic interactions (Figure 6A). Thus, subcellular variation in ion concentrations may induce fast changes in the physical properties of vimentin without affecting the keratin network [94,95]. In non-ionic buffers, the strain-stiffening behavior is more pronounced for keratin than for vimentin networks, and the G^0^ of keratin networks is higher [62]. Electrostatic repulsion by the high negative surface charge density of vimentin outbalances their weaker hydrophobic interactions. The addition of multivalent cations to the vimentin network amplifies their strain-stiffening behavior and the stress at which the network ruptures [61,63]. Controlling the composition of the network and the dynamics of binding and unbinding of inter-filament bonds may thus allow cells to adapt the resistance of their IF network and, as a consequence, cell mechanics to their environment [72].

The IF-type and ionic concentration also affect lateral filament–filament interactions responsible for bundling. K5/K14 IFs bundle readily, arranged by the K14 tail domain, which promotes strain-stiffening of the network at large deformations [75,96]. In contrast, for K8/18, the combined presence of hydrophobic interactions and electrostatic repulsion in the rod domain prevents bundling [62], which can be induced by increasing ionic concentrations [67,97]. In the case of vimentin, IF bundling is favored in high concentrations of divalent cations, which increases inter-filament electrostatic attractions [98,99]. Different ion species induce different effects; the bundling of vimentin filaments by Ca^2+^ ions is induced at two orders of magnitude higher concentrations compared to Zn^2+^ [98]. These cations bind competitively to vimentin filaments, indicating that controlling the concentrations of ion species is another mechanism for cells to tune their IF network mechanics. When lateral filament–filament interactions become too strong, they prevent other inter-filament crosslinks, leading to a decrease in network stiffness [98]. Cells may be able to change the concentration of specific ions using different types of ionic pumps to tune the organization and mechanical properties of the IF network.

Post-translational modifications [50], such as phosphorylation, can also change the charge pattern within the filament. Although the complete phosphorylation of filaments results in their disassembly, partial phosphorylation softens single filaments in vitro [100]. Different sets of experiments point towards a decrease in the lateral coupling of monomers due to phosphorylation-induced changes in charges and electrostatic interactions, leading to the softening of the filaments [100]. Interestingly, plating cells on soft substrates induces vimentin phosphorylation at pSer55 [93]. How phosphorylation changes the mechanical response of the IF network to stresses is currently unknown. Together, these studies illustrate how changes in the expression or modifications of IF proteins could modify cell mechanical properties. IFs are often heteropolymers of IF proteins, which may have distinct mechanical properties [101]. Subtle alterations of the ratio of IF protein types could provide another tool to fine-tune their mechanics. Finally, the presence of crosslinking proteins such as plectin or other yet to be identified IF-binding proteins which facilitate inter-filament interactions, could modulate inter-filament interaction dynamics and tune the mechanics of the IF network. For example, plectin has been shown to have a profound impact on cell mechanics [102,103,104], and may directly influence the intrinsic physical properties of the IF network as well. Binding of the 14-3-3 protein to IFs has been shown to alter the intrinsic physical properties of IFs and enhance their softening, leading to the hypothesis of a role for 14-3-3 in regulating IF, and thereby cell, mechanics [100].

### 5.2. Intermediate Filaments Affect the Mechanics of Cytoskeletal Composite Networks

In cells, the continuous crosstalk between IFs, actin, and microtubules [48] contributes to composite network’s mechanical behavior, which, when analyzed in vitro, differs from the behavior of their individual counterparts [105,106,107].

The stiffness of the composite network formed by addition of actin and cytoskeletal crosslinkers to vimentin without changing the total concentration of proteins is higher than the stiffness of separate vimentin or actin networks [108,109]. If the mesh size is kept constant instead of the protein concentration, the stiffness of the composite networks ranges between the stiffness of the individual networks of the same mesh size. Increasing the vimentin–actin ratio increases the elasticity of the composite network [110]. With keratins, the impact of actin is different. Actin causes a steric hindrance, which reduces the bundling of keratin IFs [111,112]. Bulk shear rheology shows that, at low strain, the viscoelastic behavior of actin and K8/18 composite networks reconstituted in vitro is intermediate between that of pure networks. When submitted to large deformations, the addition of K8/18 increases the strain-stiffening properties and the yield stress [112]. Compared to vimentin, keratin is more efficient in increasing composite network strain-stiffening [110,112].

A recent study on composite networks of vimentin and microtubules has shown that vimentin modifies the stability of the microtubule network by regulating microtubule dynamic instability [113]. Optical trapping reveals strong interactions between a single microtubule and vimentin IF. What these interactions mean for the mechanical response of a composite IF–microtubule network has yet to be determined.

Altogether, analyses of composite networks suggest that the cell mechanics results from multiple interactions between the different cytoskeletal networks and that changes in the composition and the relative quantity of each cytoskeletal component may profoundly affect the global mechanical properties of cells. The role of IFs in increasing the strain-stiffening behavior of composite networks might provide mechanical resilience which is perturbed in cells expressing disease-causing desmin and keratin mutants (EBS and cardiac myopathies). In vitro networks assembled from these mutant proteins do not strain-stiffen [74,96]. The distinct effects of specific IF proteins on the mechanical properties of actin–microtubule–IF composite networks remain to be fully characterized. They will help us understand how variations in IF composition during cell differentiation or in disease can modulate cell mechanics and adapt it to changing microenvironments or different cellular functions.

## 6. Conclusions and Future Directions

In multicellular organisms, cells are continuously submitted to physical challenges whose nature and extent depend on microenvironment mechanics, cell motility within this environment, and the external mechanical stresses applied to the tissue. To resist these challenges, maintain tissue integrity, and ensure the survival of the organism, cell mechanical properties change in time and space [114]. Accumulating evidence points towards the IF network as a key player in cell mechanics and as an adaptable system because of its regulated composition and structure. This review advocates the idea of the IF network as a tunable intracellular scaffold integrating molecular, cellular, and tissue mechanics and providing cell-type-specific mechanical properties.

The characteristic mechanical properties of single filaments and filament networks are based on the amino acid sequence and the assembly of IF proteins. Disease-causing mutations of IFs illustrate the critical role of IFs as modulators of cell mechanics. Changes in IF-type expression, PTMs, pH, ionic concentrations, and direct and indirect interactions with other cytoskeletal components control IF composition, structure, and mechanical functions and can be modified in response to extracellular and intracellular mechanical signals. For instance, in the brain IF expression increases in response to inflammation as a consequence of injury, infections, or neurodegenerative diseases [115]. The resulting IF network rearrangements might change the mechanical properties of the IF network. Future studies should focus on uncovering the link between IF network rearrangements induced by mechanical signals and how such rearrangements impact the mechanical properties of the IF network. This will increase our knowledge on how IFs are tuned to optimize cell mechanics.

Cells not only have to resist physical stresses, but they must also sense mechanical information to direct a wide range of functions such as cell migration, proliferation, differentiation, or apoptosis. The cytoskeleton plays an essential role in sensing and transmitting the mechanical properties of the microenvironment. Although this review focused on IFs as the core of a mechanically resilient intracellular scaffold, one must also consider that IFs actively participate in the mechanotransduction process [24]. While their participation in the mechanotransduction process is becoming increasingly clear, their direct mechanosensing capacity is still debated. Since IFs can resist very large stresses, the force-sensing range of IFs has the potency to be much larger than that of actin. Recently, Cten/tensin 4—a member of the tensin family known to play a role in linking integrins to the actin cytoskeleton—was shown to interact specifically with stretched keratin filaments [116]. The protein detaches from the filaments when the stretching force decreases, making keratins a force-sensing element. Moreover, increasing the stress duration reduces the dissociation rate, which indicates that the protein binding contributes to the tensile memory of the filament. How the binding of Cten to keratin further participates in the transduction of mechanical forces has to be determined. Characterizing the IF-interacting domain of Cten to keratin will open up the door to discovering new force-sensing capacities of IFs. It is tempting to speculate that the conformational changes of IF proteins upon stretching expose numerous binding sides, which may differ depending on the composition of IFs. IF protein expression may thus not only control the mechanical properties of the cells but also modify their ability to transduce different ranges of mechanical cues as well as the cellular responses induced by these cues.

In conclusion, the unique mechanical properties of IFs, in combination with their high degree of tunability and adjustability to the mechanical needs of tissues and cells, make them key players in integrating mechanical information to support and direct cell and tissue function.

## Figures and Tables

**Figure 1 cells-10-01905-f001:**
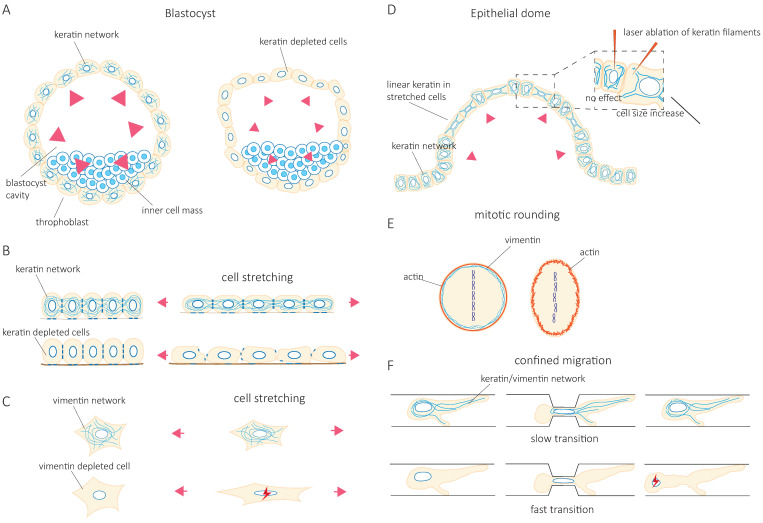
Intermediate filaments, guardians of tissue and cell integrity, adapt cell mechanics to cell behavior. (**A**). During embryo cavitation, keratins are essential to generate apical tension (pink arrowheads) against the increasing internal pressure in the blastocyst. This tension is lost in the absence of keratin 8 and 18, which leads to a decrease in volume and increased surface curvature. (**B**,**C**). Depletion of keratin in a collective cell sheet and vimentin in single cells increases cell deformation upon stretching. In vimentin-depleted cells, this leads to an increase in cell death. (**D**). Expanded epithelial domes consist of stretched and unstretched cells. Stretched cells contain unusually straight keratin bundles, which, when disrupted by laser ablation, cause the cell to lose its shape and largely increase its area. (**E**). The vimentin network contributes to cortical tension during mitotic rounding. The loss of vimentin impairs rounding and induces abnormalities in chromosomal aggregation. (**F**). Both keratin and vimentin IFs slow down confined migration. Depletion of keratin or vimentin in different cell types increases their confined migration speed but promotes nuclear damage (red lightning sign).

**Figure 2 cells-10-01905-f002:**
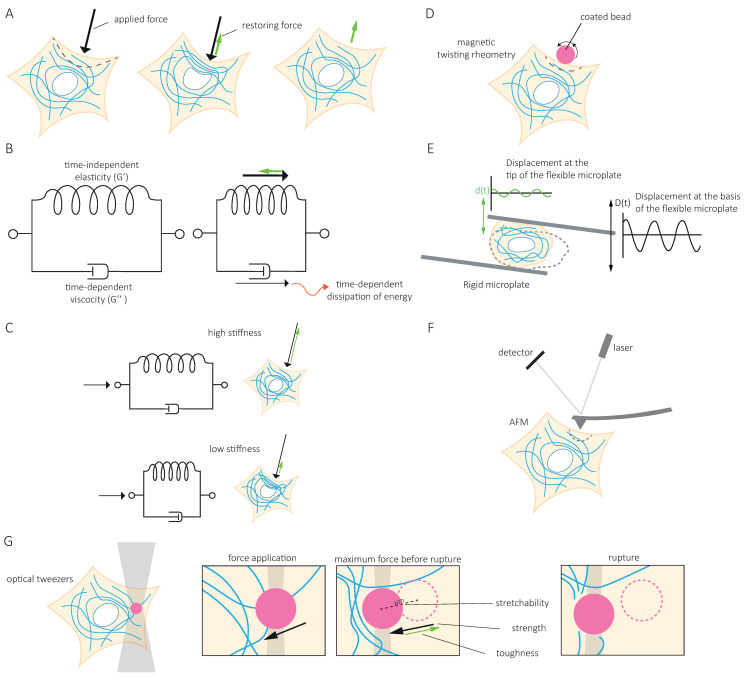
Intermediate filaments and cell-intrinsic mechanical properties. (**A**). When a force is applied to a cell (black arrow), part of the energy used to induce cell deformation (grey dashed line) is stored. Once the force is no longer applied, the stored energy is used to generate a restoring force (green arrow) which reduces (or abolishes) the deformation. (**B**). The deformability of the cell and its ability to return to its original shape depend on its viscoelastic properties. A cell both behaves as an elastic solid and a viscous fluid. Elasticity makes a cell resistant to deformations, comparable to a spring, where the energy that induces the deformation is stored in the material independent of time. Viscosity makes material flow like a fluid (represented as a damper) where the material’s resistance depends on the rate of deformation, and the energy that is put into the deformation is not stored in the material but is dissipated. The elastic modulus is indicated by G’ and the viscous modulus by G’’ in Pa. (**C**). To indicate the overall resistance of a cell against deformation, the overarching term stiffness is used. Stiffness does not distinguish between the elastic and viscous properties of the material. (**D**). Twisting magnetic beads coated with different proteins which interact with the cell surface can be used to deform a cell and determine its resistance to deformation, e.g., stiffness. (**E**). Single-cell rheology, in which cells are deformed in between a rigid and a flexible microplate, can be used to determine cell stiffness from the displacement of the flexible microplates. (**F**). Atomic force microscopy determines the stiffness of cells by locally probing the surface of the cell using a cantilever which contacts the surface. The bending of the cantilever reflects the stiffness of the cell and is determined by the reflection of laser light from the cantilever detected by a photodiode detector. (**G**). Intracellular magnetic beads larger than the mesh size of the cytoskeleton can be controlled using optical tweezers. Dragging intracellular beads at a constant speed towards the nucleus increasingly deforms the cell, and the energy needed to reshape the cell after the force is released is stored (green arrow). In cells, the amount of energy that can be stored has a maximum, after which, the material yields and changes its behavior from elastic to plastic. This maximum, or peak force, defined as cytoplasmic strength, indicates how much force can be applied to the cell until it cannot recover from its deformation. The distance the bead can move until this peak force is reached is defined as stretchability (µm). The sum of all the energy the cell can take up is defined as cytoplasmic toughness.

**Figure 3 cells-10-01905-f003:**
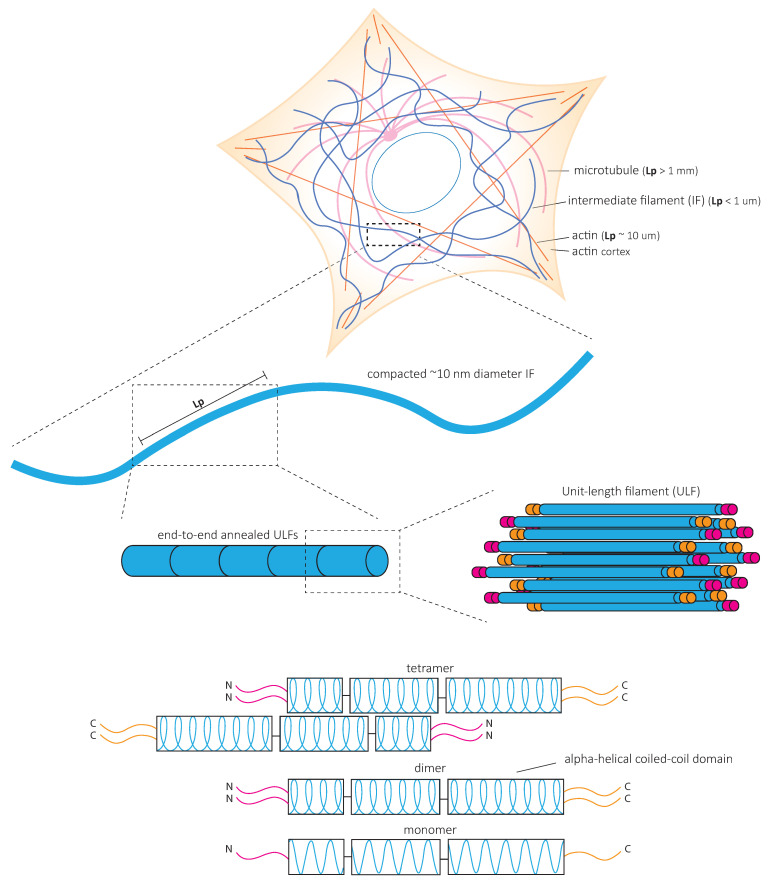
Intermediate filament structure. All intermediate filament (IF) proteins are formed by an α-helical ‘rod’ domain flanked by unstructured N- and C-termini that extend out of the assembled filament. The rod domain contains three coiled-coil domains responsible for the formation of IF dimers [51]. Dimers assemble in an anti-parallel fashion into apolar tetramers. Tetramers assemble via lateral associations into a unit length fragment (ULF) that anneal longitudinally and eventually form long 10 nm diameter filaments after lateral compaction. Soluble IF tetramers contribute to subunit exchange [52,53,54,55,56,57] and may influence the stability of the filaments and the mechanical properties of the network [57]. Lp = persistence length.

**Figure 4 cells-10-01905-f004:**
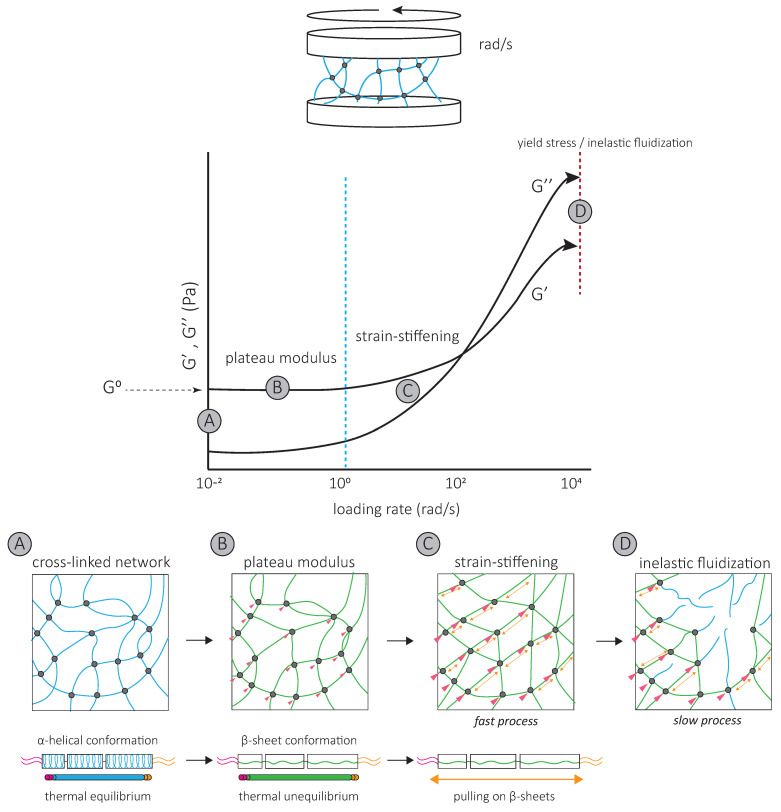
The mechanical properties of IF networks characterized by oscillatory shear rheology experiments of in vitro assembled networks (**A**). Deformation of the network using oscillatory shear rheology is achieved by network assembly between two plates and rotating one plate while the other is fixed, inducing a controlled deformation. Increasing the frequency of the rotations (rad/s) increases the stress and strain applied to the network. The response of the IF network to oscillatory shear is viscoelastic. Within the range of 0.01 and 10 rad/s, G’ and G’’ are independent of the oscillatory frequency, and a plateau is observed in both the G’ and G’’ curves. The corresponding G’ is called the plateau modulus or G^0^ (**B**). Further increasing the stress by increasing the oscillatory shear frequency results in strain-stiffening of the network ((**C**), start is indicated with blue dotted line). The network strain-stiffens up to a critical stress, the yield stress ((**D**), red dotted line), at which the network ruptures. This corresponds to the start of inelastic fluidization. The plateau modulus results from attractive forces between filaments (grey circles) to maintain the stretched β-sheet conformation of single filaments between crosslinks which are in thermal unequilibrium ((**B**), green filaments). The weaker inter-filament interactions between filament rod domains are thought to be responsible for the attractive interactions at low loading rates. Lower forces are required to resist the lower stresses (pink). Strain-stiffening results from the stretching of single filaments in β-sheet conformations ((**C**), yellow arrows) maintained by stronger inter-filament interactions. C-terminal tails of IFs are thought to be responsible for these inter-filament attractive interactions, which have to resist high stresses (pink). At the critical yield stress, the network ruptures as a result of the unbinding of multiple crosslinks (**D**). Strain-stiffening and inelastic fluidization compete at high stresses, resulting in a loading-rate-dependent rupture of the network [72]. Strain-stiffening is a fast process (**C**) while inelastic fluidization is slow (**D**). At fast loading rates, strain-stiffening dominates, and the network can resist higher stresses. At slow loading rates, the unbinding of crosslinks has time to occur within the same time frame as strain-stiffening, resulting in rupture of the network at lower stresses.

**Figure 5 cells-10-01905-f005:**
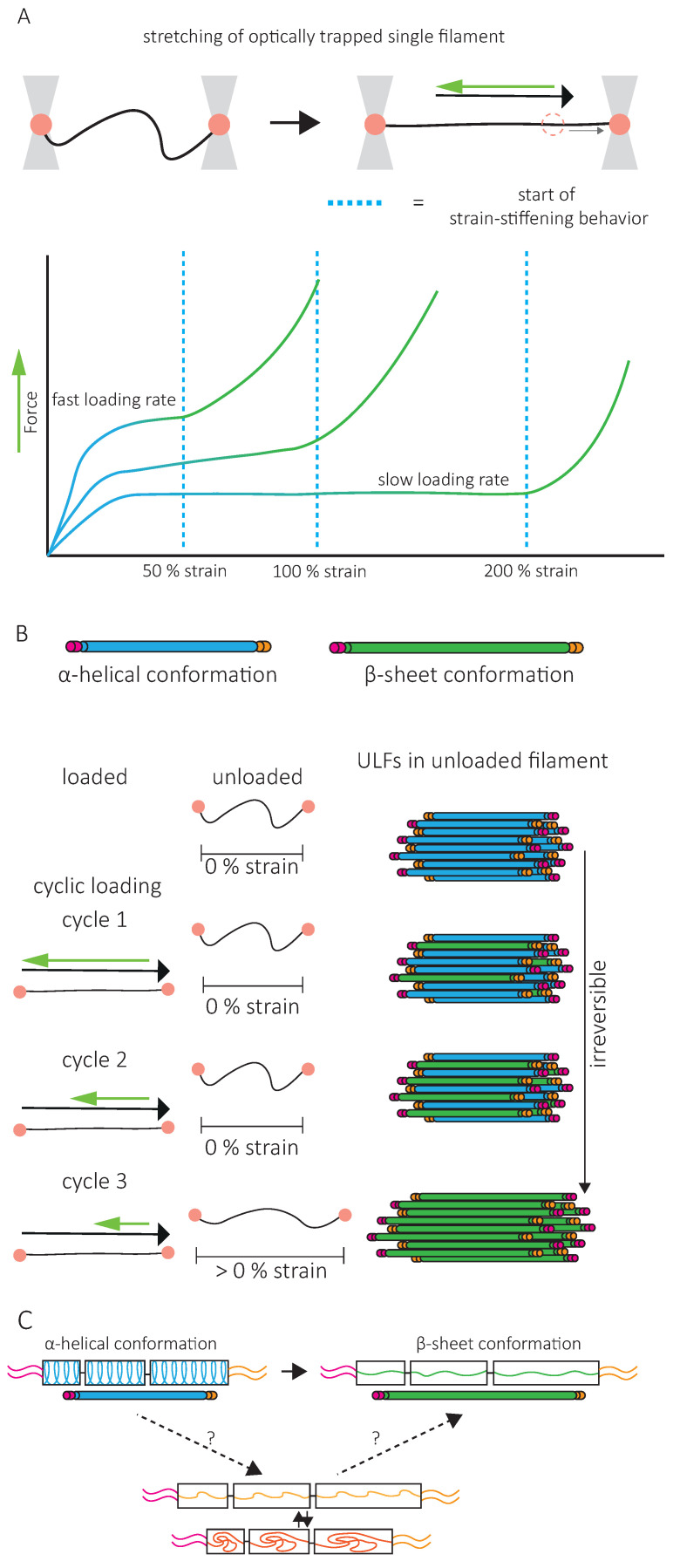
Single intermediate filament mechanics and underlying conformational changes. (**A**). Optical trap of a single intermediate filament and force–strain curves generated from stretching a single filament at different loading rates. The curve is characterized by a steep increase in force at low strains derived from the stretching of the alpha-helices (blue line and dimer). This is followed by a plateau in the curve at intermediate strains, attributed to the unfolding of α-helices to β-sheet conformations (green line and dimer). At higher strains, the filament strain-stiffens as a result of pulling on β-sheets. Both filaments stretched by atomic force microscopy and optical trapping demonstrate this mechanical response. For fast loading rates, strain-stiffening is already observed at 50% strain, while at slow loading rates, strain-stiffening is not observed before 200% strain. (**B**). For repeated loading and unloading cycles, the force needed to reach the same strain decreases with each cycle. This is attributed to part of the dimers being in a β-sheet conformation (green) while others remain in an α-helical conformation (blue). Lower forces are required to unfold the remaining α-helices. The filament length after the loading force is released is determined by the shortest elements, thus α-helices. After all α-helices have switched to a β-sheet conformation, elongation of the unloaded filament can be observed. Experiments have shown that the original mechanical properties are irreversible within physiologically relevant recovery times. The β-sheet conformation requires energy and is in thermal unequilibrium. Upon unloading, the filament is expected to return to its α-helical thermal equilibrium conformation. However, the irreversibility of its original mechanical properties does not fit this explanation. (**C**). To explain the irreversibility of the original mechanical properties of single filaments, a third disordered conformational state has been proposed (orange), which can elongate (yellow) but cannot return back to an α-helical conformation. Adapted from Block et al., Phys. Rev. Lett 2017 & Forsting et al., Nano Letters 2019.

**Figure 6 cells-10-01905-f006:**
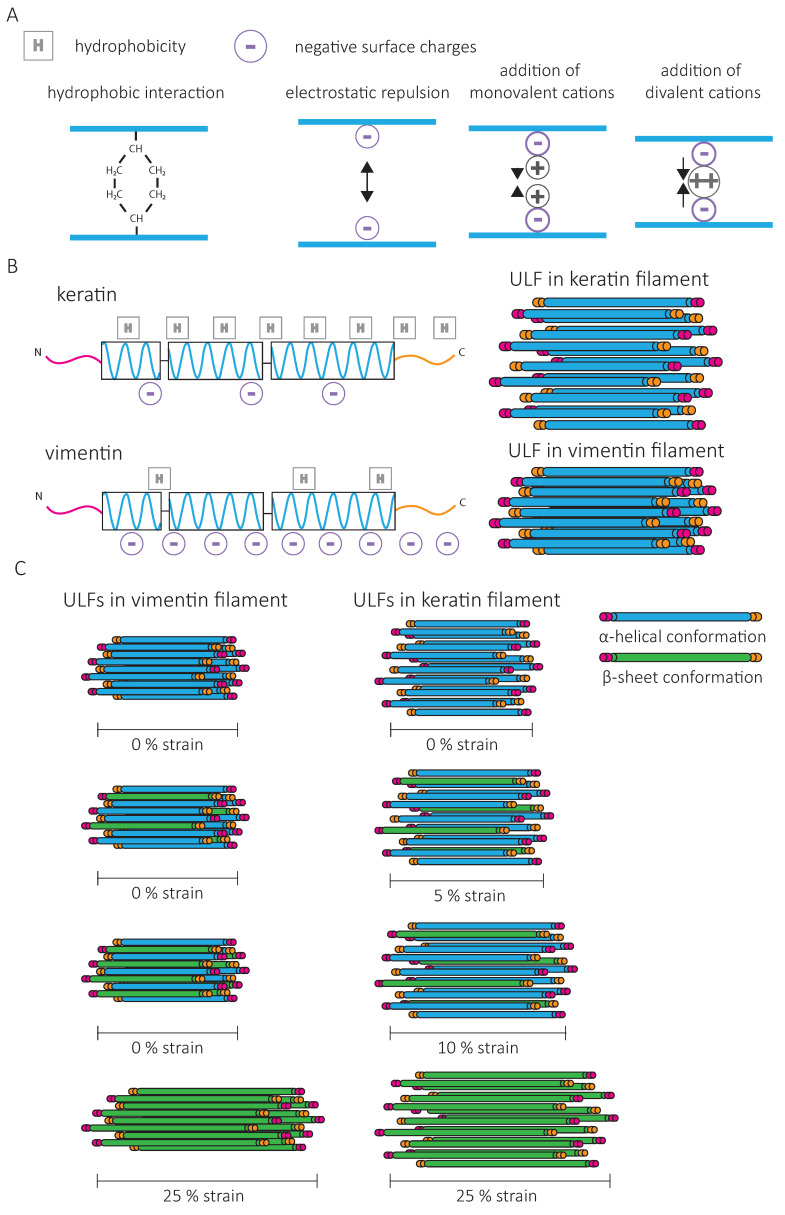
IF subunit interactions and diversity. (**A**). Interactions between intermediate filament proteins result from hydrophobic (H) and electrostatic interactions (circled charges). The presence of cations can tune electrostatic interactions. (**B**). Differences in amino acid sequences between vimentin and keratin account for differences in hydrophobic and electrostatic interactions. Lateral coupling within single vimentin IFs is much more pronounced compared to keratin IFs in the presence of high ionic concentrations due to higher negative surface charges. (**C**). Differences in hydrophobic and electrostatic interactions account for the different mechanical responses observed for vimentin and keratin. Increased lateral coupling between vimentin filaments makes elongation impossible unless all subunits have adopted the β-sheet conformation (green). Keratin filaments elongate as soon as some subunits are in a β-sheet conformation. ULF = Unit Length Fragment. Adapted from Lorenz et al., Phys. Rev. Lett. 2019.

## Data Availability

Not applicable.

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
