# Peer review of "Intermediate Filaments from Tissue Integrity to Single Molecule Mechanics"

_cells, 2021, doi:10.3390/cells10081905_

Round 1

Reviewer 1 Report

This is a well-written review. I have no major and minor concerns.

Author Response

Answers to the reviewer 1’s comments

This is a well-written review. I have no major and minor concerns.

Response:

We thank the reviewer for the kind words.

Reviewer 2 Report

The manuscript entitled ‘Intermediate filaments from tissue integrity to single molecule mechanics’ by Emma J. van Bodegraven and Sandrine Etienne-Mannevilleet represents a very interesting review manuscript where the authors summarized the current knowledge regarding the contribution of IFs to cell- and tissue mechanics. Further, the authors also summarized the in vitro studies where the physical properties of single IFs and IF networks were explored. Finally, the authors referred some factors that can fine tune IFs properties to adapt cell and tissue mechanics according the environmental changes.

In overall, I consider that the premise of this study is very interesting and important for the field, and I will perform some comments and suggestions.

Minor concerns:

  1. Some statements could to be improved/clarified:
  • ‘These characteristics are probably responsible for our late understanding of IF functions at the cellular and tissular level’ (lines 47-49). Tissular? Do you mean tissue? If so, replace the term all over the manuscript.
  • Please clarify the following statement: ‘The critical function of keratins in tissue integrity was recently demonstrated during early development when asymmetric inheritance of keratin drives the specification of the placenta and the embryo’ (lines 64-65).
  • Please clarify the following statement: ‘These observations further emphasize the need for context-dependent investigation of the adaptive and active role of IFs in cell mechanics’ (lines 117-118).
  • Please clarify the following statement: ‘The filament persistence length gives us access to their bending rigidity, which quantifies the flexibility of the filament’ (lines 303-304).
  • Please clarify the following statement: ‘Although the mechanisms by which IF mechanics can be tuned are just starting to be uncovered, changes in IF-type expression, PTMs, pH, ionic concentrations, and direct and indirect interactions with other cytoskeletal components are key parameters affecting IF network mechanics’ (lines 478-481). Further, the future directions should be clearly pointed by the authors.

  1. The images presented by the authors are very elucidative, however are very large and should be improved. One page per image is too large! Some information is missing (is cut). Always use the same organization in all Figures, A, B, C… In figure 1, the authors label A,B,C in left and D, E, F in the right. The figure 2 was labelled in a different way!

  1. The sections should be labelled only with numbers: 2, 2.1,…and 3, 3.1 , 3.2,…and not numbers and letters!The title of topic 4 should be improved: ‘In vitro mechanical properties of single IFs and IF networks’. Other headings could be improved.

  1. Replace the in vitro by the italic version, all over the manuscript.

Author Response

Answers to the reviewer 2’s comments

The manuscript entitled ‘Intermediate filaments from tissue integrity to single molecule mechanics’ by Emma J. van Bodegraven and Sandrine Etienne-Mannevilleet represents a very interesting review manuscript where the authors summarized the current knowledge regarding the contribution of IFs to cell- and tissue mechanics. Further, the authors also summarized the in vitro studies where the physical properties of single IFs and IF networks were explored. Finally, the authors referred some factors that can fine tune IFs properties to adapt cell and tissue mechanics according the environmental changes.

In overall, I consider that the premise of this study is very interesting and important for the field, and I will perform some comments and suggestions.

We thank the reviewer2 for the kind words and the comments which we have used to improve our manuscript. Here is our point-by-point answer to the reviewer’s comments:

  1. Some statements could to be improved/clarified:
  • ‘These characteristics are probably responsible for our late understanding of IF functions at the cellular and tissular level’ (lines 47-49). Tissular? Do you mean tissue? If so, replace the term all over the manuscript.

We have changed ‘tissular’ to ‘tissue’ throughout the manuscript.

  • Please clarify the following statement: ‘The critical function of keratins in tissue integrity was recently demonstrated during early development when asymmetric inheritance of keratin drives the specification of the placenta and the embryo’ (lines 64-65).

We have modified this sentence.

  • Please clarify the following statement: ‘These observations further emphasize the need for context-dependent investigation of the adaptive and active role of IFs in cell mechanics’ (lines 117-118).

We have modified this sentence.

  • Please clarify the following statement: ‘The filament persistence length gives us access to their bending rigidity, which quantifies the flexibility of the filament’ (lines 303-304).

We have changed this sentence to : ‘The flexibility of filaments is quantitatively characterized by their bending rigidity. The bending rigidity can be derived from the filament persistence length, which is the length over which the filament direction does not change (Figure 3).’ and have indicated the persistence length in Figure 3.

  • Please clarify the following statement: ‘Although the mechanisms by which IF mechanics can be tuned are just starting to be uncovered, changes in IF-type expression, PTMs, pH, ionic concentrations, and direct and indirect interactions with other cytoskeletal components are key parameters affecting IF network mechanics’ (lines 478-481). Further, the future directions should be clearly pointed by the authors.

We have changed the sentence and added a short paragraph: “Changes in IF-type expression, PTMs, pH, ionic concentrations, and direct and indirect interactions with other cytoskeletal components control IF composition, structure, and mechanical functions and can be modified in response to extracellular and intracellular mechanical signals. For instance, in the brain IF expression increases in response to inflammation as a consequence of injury, infections, or neurodegenerative diseases [106]. The resulting IF network rearrangements might change the mechanical properties of the IF network. Future studies should focus on uncovering the link between IF network rearrangements induced by mechanical signals and how such rearrangements impacts the mechanical properties of the IF network. This will increase our knowledge on how IFs are be tuned to optimize cell mechanics.”

2.. The images presented by the authors are very elucidative, however are very large and should be improved. One page per image is too large! Some information is missing (is cut). Always use the same organization in all Figures, A, B, C… In figure 1, the authors label A,B,C in left and D, E, F in the right. The figure 2 was labelled in a different way!

 We have compressed some of the figures (Figure 1, 2, 5 and 6) and have changed the labelling in Figure 2 to maintain the same organization of the labelling throughout the manuscript as much as possible.

  1. The sections should be labelled only with numbers: 2, 2.1,…and 3, 3.1 , 3.2,…and not numbers and letters!The title of topic 4 should be improved: ‘In vitro mechanical properties of single IFs and IF networks’. Other headings could be improved.

We have changed the section labels accordingly and tried to improve the headings

  1. Replace the in vitro by the italic version, all over the manuscript.

We have replaced ‘in vitro’ by ‘in vitro’ throughout the manuscript.

Reviewer 3 Report

In their manuscript “Intermediate filaments from tissue integrity to single molecule mechanics” Bodegraven and Etienne-Manneville reviewed the role of intermediate filaments (IFs) in cell and tissue mechanics in the context of recent in vitro studies. The manuscript addresses a very interesting and topical field. Although focus of review is clearly on in vitro data, authors should significantly strengthen part covering in vivo experiments (especially if they intended to cover the topic “from tissue to single molecule”). Authors also missed opportunity to discuss in more detail relevance (or lack of relevance) of in vitro findings for conclusions made for cells/tissues. The manuscript is acceptably well structured and contains a lot of literature reflecting the current state of IF research. The authors provide an informative overview. Nevertheless, I have some additional points concerning the current version, which should be rectified before review acceptance:

1) Lines 143-153 and Figure 2: Might be better and more useful if the information is more structured and a chapter describing techniques to study biophysical properties of IFs is included. Separation of method description and a review of IFs biophysical properties will be very desirable within whole chapter 3.

2) Chapter 4 requires better, more detail introduction to allow broader readership to fully understand and follow the text.

3) Fig. 1D: This is misleading schematics, as in the original publication (Latorre et al 2018) authors show that cells at the top of the dome are stretched, while those at the sides of the dome are not. This schematic shows rather random localization of stretched cells within the epithelial dome.

4) Line 110-112: “In HeLa Kyoto cells, vimentin IFs contribute to this increase in cortical tension by relocalizing to the cortex, where they interact with actin to control actin organization [28,29] (Figure 1E).” Here authors do not discuss obvious weakness of cited paper - HeLa cells have besides vimentin IFs also keratins and previous works by Werner Franke demonstrated that, though the keratin network reorganizes strongly during mitosis, it is still there in various types of structures.

5) Line 79-89: Ref. 16 describes rather a unique/unusual finding (in the context of other publications), as vimentin depletion is usually associated with decreased cell stiffness and associated increased deformability during various physiological processes such as migration through confined spaces or mechanical resilience during mitosis. Relevant contradicting papers should be cited here (Patteson 2019, PMID: 31721440; Serres 2020, PMID: 31928973; Patteson 2019, PMID: 31676718 and others...). Any citations to back up claim: “For example, depletion of vimentin in compressed highly contractile cells might have a different effect on their deformability compared to depletion in cells with lower contractility.”?

6) Line 432-434 “Finally, crosslinking proteins such as plectin or other yet to be identified IF-binding proteins can also modulate inter-filament interactions and tune the mechanics of the IF network.” This is gross understatement as crosslinkers (such as mentioned plectin) do not modulate but facilitate inter-filament interactions and are well established in controlling the mechanics of the IF networks and by extension also cells/tissues. Relevant citations have to be included (e.g. Osmanagic-Myers 2006, PMID: 16908671; Na 2009, PMID: 19244477; Jirouskova 2018, PMID: 29273475).

Minor points:

1) In the headlines, sometimes “Intermediate Filaments” are written with both I and F capitalized, somewhere not, as “Intermediate filaments” – should be unified (preferably not capitalized).

2) Usage of “IF network” and “IF networks” is inconsistent. Please revise.

3) Line 291-292: At fast deformations, these interactions provide IF network with (?) resilience [69].

4) Line 641: 2006 should be in bold.

5) Line 354: „in vitro“ should be in italic

Author Response

Answers to the reviewer 3’s comments

In their manuscript “Intermediate filaments from tissue integrity to single molecule mechanics” Bodegraven and Etienne-Manneville reviewed the role of intermediate filaments (IFs) in cell and tissue mechanics in the context of recent in vitro studies. The manuscript addresses a very interesting and topical field. Although focus of review is clearly on in vitro data, authors should significantly strengthen part covering in vivo experiments (especially if they intended to cover the topic “from tissue to single molecule”). Authors also missed opportunity to discuss in more detail relevance (or lack of relevance) of in vitro findings for conclusions made for cells/tissues. The manuscript is acceptably well structured and contains a lot of literature reflecting the current state of IF research. The authors provide an informative overview. Nevertheless, I have some additional points concerning the current version, which should be rectified before review acceptance.

We thank the reviewer for his encouraging and constructive comments. Here our point-by-point answers to his/her comments.

  • Lines 143-153 and Figure 2: Might be better and more useful if the information is more structured and a chapter describing techniques to study biophysical properties of IFs is included. Separation of method description and a review of IFs biophysical properties will be very desirable within whole chapter 3.

We have structured the first paragraph of Chapter 3 by describing the different techniques which will be used later on to discuss the role of IFs in the mechanical properties of cells. In the introduction, we now refer to Figure 2 where the biophysical approaches are explained in greater detail.

  • Chapter 4 requires better, more detail introduction to allow broader readership to fully understand and follow the text.

We have added a couple of sentences to detail the different aspects of chapter 4 in the first paragraph.      

  • 1D: This is misleading schematics, as in the original publication (Latorre et al 2018) authors show that cells at the top of the dome are stretched, while those at the sides of the dome are not. This schematic shows rather random localization of stretched cells within the epithelial dome.

We thank the reviewer for pointing out the discrepancy between our schematic and the original figures. We have adjusted Figure 3 and localized the highly stretched cells to the top of the domes as depicted in the original paper and images of the domes.

  • Line 110-112: “In HeLa Kyoto cells, vimentin IFs contribute to this increase in cortical tension by relocalizing to the cortex, where they interact with actin to control actin organization [28,29] (Figure 1E).” Here authors do not discuss obvious weakness of cited paper - HeLa cells have besides vimentin IFs also keratins and previous works by Werner Franke demonstrated that, though the keratin network reorganizes strongly during mitosis, it is still there in various types of structures.

We have clarified our statement in which we now point out that it is important to take into account the other IF proteins expressed in HeLa cells.

  • Line 79-89: Ref. 16 describes rather a unique/unusual finding (in the context of other publications), as vimentin depletion is usually associated with decreased cell stiffness and associated increased deformability during various physiological processes such as migration through confined spaces or mechanical resilience during mitosis. Relevant contradicting papers should be cited here (Patteson 2019, PMID: 31721440; Serres 2020, PMID: 31928973; Patteson 2019, PMID: 31676718 and others...). Any citations to back up claim: “For example, depletion of vimentin in compressed highly contractile cells might have a different effect on their deformability compared to depletion in cells with lower contractility.”?

We agree with the reviewer that the findings published in reference [16] are contradicting to the studies mentioned by the reviewer. This is why we have given several explanations for this contradiction which require to be investigated in future studies and therefore are currently not backed up by any experimental data. The references mentioned by the reviewer are discussed in Chapter 2.3, since the changes in mechanical properties observed in these studies are specifically related to a certain cell behavior such as migration (Patteson 2019, PMID: 31721440; Patteson 2019, PMID: 31676718) and cell division (Serres 2020, PMID: 31928973).

  • Line 432-434 “Finally, crosslinking proteins such as plectin or other yet to be identified IF-binding proteins can also modulate inter-filament interactions and tune the mechanics of the IF network.” This is gross understatement as crosslinkers (such as mentioned plectin) do not modulate but facilitate inter-filament interactions and are well established in controlling the mechanics of the IF networks and by extension also cells/tissues. Relevant citations have to be included (e.g. Osmanagic-Myers 2006, PMID: 16908671; Na 2009, PMID: 19244477; Jirouskova 2018, PMID: 29273475).

We have adjusted our statement on the crosslinking influence on the mechanical properties of IF network. We now also discuss the involvement of plectin to the intrinsic IF network mechanics in more detail by including the references in which the role of plectin in cell mechanics was addressed, as suggested by the reviewer.

Minor points:

1) In the headlines, sometimes “Intermediate Filaments” are written with both I and F capitalized, somewhere not, as “Intermediate filaments” – should be unified (preferably not capitalized).

2) Usage of “IF network” and “IF networks” is inconsistent. Please revise.

3) Line 291-292: At fast deformations, these interactions provide IF network with (?) resilience [69].

4) Line 641: 2006 should be in bold.

5) Line 354: „in vitro“ should be in italic

We thank the reviewer for pointing out minor points to improve the manuscript. We have adjusted them according to the suggestions. As for the point 2, we have consistently used ‘IF networks’ when several distinct networks of IFs are involved.

Round 2

Reviewer 3 Report

The authors have addressed my previous concerns. No further revision is required. Please check once again for spelling and typos, e.g.:

Line 462 which facilitatae inter-filament interactions, vould